# Microbiological Characterization of Protected Designation of Origin Serra da Estrela Cheese

**DOI:** 10.3390/foods12102008

**Published:** 2023-05-16

**Authors:** Rui Rocha, Nélson Couto, Ricardo Pereira Pinto, Manuela Vaz-Velho, Paulo Fernandes, Joana Santos

**Affiliations:** 1CISAS—Center for Research and Development in Agrifood Systems and Sustainability, Instituto Politécnico de Viana do Castelo, Rua Escola Industrial e Comercial de Nun’Álvares, 4900-347 Viana do Castelo, Portugal; rru@ipvc.pt (R.R.); rpinto@ipvc.pt (R.P.P.); paulof@estg.ipvc.pt (P.F.); joana@estg.ipvc.pt (J.S.); 2Escola Superior de Tecnologia e Gestão, Instituto Politécnico de Viana do Castelo, Avenida do Atlântico no. 644, 4900-348 Viana do Castelo, Portugal; nelsoncouto@ipvc.pt

**Keywords:** traditional Portuguese cheeses, Serra da Estrela, protected designation of origin, lactic acid bacteria, food safety/hygienic indicators, ewe raw milk, cardoon (*Cynara cardunculus* L.)

## Abstract

Serra da Estrela is the oldest and most recognizable traditional protected designation of origin (PDO) cheese from Portugal. It has been extensively studied over the years, but the latest microbial characterization is 20 years old. Hence, this work aimed to perform an updated characterization of Serra da Estrela PDO cheeses and raw materials. Our analysis showed that lactic acid bacteria content on Serra da Estrela cheeses exceeded 8.8 log CFUsg^−1^, in all analyzed samples, with lactococci, lactobacilli and *Leuconostoc* spp. predominating over enterococci strains. Moreover, lactococci and lactobacilli abundance increased across the production season, while enterococci dropped considerably in late manufactures. Lastly, *Leuconostoc* spp. content remained unchanged in all analyzed periods. A correspondence analysis showed that *L. paracasei*, *L. lactis*, *E. durans*, *E. faecium* and *L. mesenteroides* are transversal in Serra da Estrela cheese manufacturing and were closely associated with milk, curd and cheese matrices. Additionally, *L. casei*, *L. plantarum* and *L. curvatus* were specifically associated with cheese matrices, possibly active during ripening and contributing for the development of these cheeses’ organoleptic characteristics.

## 1. Introduction

Cheese making is a millenary art, tracing back to the first agricultural revolution, where humankind took the first steps from a hunter-gatherer to sedentary societies [1]. Its more basic form consists of a biochemically driven process of milk coagulation into a soft gel that is filtered, and the resulting curd is allowed to ripen for a certain period until the desired characteristics are obtained [2,3].

Cheese is the most diverse and difficult to classify foodstuff among dairy products. To this date, there is no definitive list of cheese varieties, and some authors suggest there are over 1000 varieties and/or variants, most of them of artisanal manufacturing. This diversity results from the wide variety of elemental and/or methodological approaches in cheese making that ultimately lead to distinctive visual, textural, physical, aromatic and flavor attributes. Elemental variation is linked to the type of milk (with or without heat treatment [raw] from ewe, goat, cow or others), coagulant (animal, vegetal or synthetic rennet) and starter and/or adjunct cultures used. Additionally, methodological variation arises from divergent approaches to milk setting, cutting, stirring, heating, draining, pressing, salting and ripening steps [1,4,5,6].

Cheese manufacturing is a complex biochemical and biological process, with a highly selective and dynamic three−dimensional microbial ecosystem of bacteria, yeasts and/or molds, which are ultimately responsible for the development of the organoleptic characteristics found in each type of cheese [1,4,7,8,9,10]. In fact, artisanal cheeses manufactured following traditional procedures in small-scale dairy farms with low to no mechanization are, in comparison to industrialized large-scale productions, generally perceived by consumers to yield cheeses of superior quality [8]. A key factor in the abovementioned cheese-making methods is the use of raw unprocessed milk that harbors a rich microbiota, whose metabolic interaction during cheese manufacturing leads to shorter ripening periods and the production of large amounts of volatile and non-volatile compounds. This produces organoleptic-rich and complex cheeses packed with flavors and aromas [8,11,12,13].

Despite the presence of a beneficial microbial consortia, raw milk can also harbor a series of pathogenic microorganisms that can, under the right circumstances, persist in raw milk cheeses. These include *Bacillus cereus*, *Clostridium botulinum*, enteropathogenic *Escherichia coli*, *Listeria monocytogenes*, *Salmonella* spp., *Staphylococcus aureus* and others [11,12,14]. Pathogenic bacteria can also gain access to cheese during manufacturing due to faulty sanitary and/or hygiene practices [15]. In fact, in the last 40 years, over 60 food-borne outbreaks were reported worldwide linked to the consumption of contaminated cheese products [14,16]. In an effort to reduce risk, legislation has been strengthened, including the introduction of the Hazard Analysis and Critical Control Points (HACCP) systems and systematic microbiological controls throughout the supply chain [8,14].

Serra da Estrela is the oldest and most renowned traditional cheese from Portugal, and it has been granted the protected designation of origin (PDO) status since 1996 [17]. Its microbiological, biochemical and sensorial characteristics have been extensively studied (reviewed by Inácio et al. [18]). However, most of microbiological analyses available date back to the end of the 20th century. Thus, this work aims to perform un updated evaluation of the microbial community associated with Serra da Estrela PDO cheeses and raw materials across the production campaign focused on viable lactic acid bacteria (LAB) content assessments. Moreover, a statistical methodology—correspondence analysis—will be used on matrix-derived genetically identified LAB isolates data. This will assist in the identification of core strains from Serra da Estrela PDO cheeses and raw materials, which could, in conjugation with other methods, contribute to the development of an autochthonous set of starter, secondary and adjunct cultures.

## 2. Materials and Methods

### 2.1. Sampling Strategy and Cheese Manufacture

Microbial characterizations were performed on ewe raw milk, cardoon, curd and cheese samples obtained in (1) November–January; (2) February–March and (3) May–June periods within the 2018/2019 production campaign (Figure 1). Thus, reflecting distinct weather and pasture conditions that were shown to influence the overall microbial composition of Serra da Estrela cheeses in previous reports [19,20,21]. Moreover, it allowed an assessment of early, mid and late manufactures, considering that artisanal Serra da Estrela cheese manufacturing elapses from November until July [22].

The Serra da Estrela cheeses were manufactured in a certified PDO cheese producer from Gouveia, Portugal. Milk and cardoon samples were collected immediately before use, while curd samples were collected prior to the molding stage. Serra da Estrela cheeses were produced following standard procedures and a ripening period of 38 days. All samples were transported and kept under refrigeration (≤4 °C) before processing.

### 2.2. Microbial Characterization Analysis

The viable microbial load of Serra da Estrela cheese, curd, milk and cardoon samples were analyzed by culture plating and colony counting into a selection of non-selective, selective, elective and differential media. To that end, 25 g of cheese, curd and milk samples were aseptically transferred to a sterile homogenization bag with a strainer and homogenized in 225 mL of buffered peptone water (Liofilchem srl, Roseto degli Abruzzi, Italy) using a paddle blender (Seward Ltd., West Sussex, UK) for 1 min. Cardoon samples were homogenized as previously reported by Rocha and co-workers [23]. Briefly, 8–10 g of dried thistle flower samples were suspended in 100 mL of a sterile solution containing 10% (*v*/*v*) buffered peptone water (Liofilchem srl, Roseto degli Abruzzi, Italy) and 0.01% (*v*/*v*) Tween 80^®^ (PanReac AppliChem, Barcelona, Spain). Subsequently, these suspensions were subjected to an ultrasonic bath for 30 s (Jet Program option) (Soltec, Milan, Italy) followed by 150 rpm orbital agitation for 30 min at room temperature (B. Braun Biotech International, Melsungen, Germany). Finally, they were poured into a homogenization bag with a strainer.

Serial decimal dilutions in maximum recovery diluent (Liofilchem srl, Roseto degli Abruzzi, Italy) were prepared with the filtered homogenates and used for microbial characterization. To that end, 100 µL of bacterial suspension dilutions were inoculated in duplicate for the enumeration of presumptive: (1) total lactic acid bacteria on Man, Rogosa and Sharpe Agar (MRSA [VWR International, Leuven, Belgium]); (2) lactococci on M17 Agar (Liofilchem srl, Roseto degli Abruzzi, Italy); (3) lactobacilli on Rogosa Agar (RA [Biokar Diagnostics, Beauvais, France]); (4) enterococci on Slanetz Bartley Agar (SBA [VWR International, Leuven, Belgium]); (5) *Leuconostoc* spp. on Mayeux, Sandine and Elliker Agar (MSE [Biokar Diagnostics, Beauvais, France)] and (6) psychrophilic microbiota on Nutrient Agar (NA [Biokar Diagnostics, Beauvais, France]). The inoculated agar plates of MRSA, M17 and RA were incubated anaerobically (Thermo Scientific, Basinstoke, UK), while SBA, MSE, NA and a second set of MRSA plates were incubated aerobically at 30 °C for 48 to 72 h, except for SBA plates that were incubated at 37 °C and NA plates incubated at 10 °C for 10 days. The grown colonies were sorted according to their phenotypic characteristics, and three representative LAB specimens were isolated onto MRSA plates for biochemical confirmation, including Gram-staining and catalase (Sigma-Aldrich GmbH, Steinheim, Germany) and cytochrome oxidase activity (Merk KgaA, Darmstadt, Germany) tests. Milk, cardoon, curd and cheese samples were also screened for the enumeration of (1) β-glucuronidase-positive *Escherichia coli* according to ISO 16649-2:2001 [24]; (2) *Enterobacteriaceae* according to ISO 21528-2:2017 [25]; (3) *Bacillus cereus* according to ISO 7932:2004 [26]; (4) coagulase-positive staphylococci according to ISO 6888-1:1999 [27]; (5) *Listeria monocytogenes* and *Listeria* spp. according to ISO 11290-2:2017 [28] and (6) yeasts and molds according to ISO 21527-1:2008 [29]. Milk, curd and cheese samples were screened for the detection of *Salmonella* spp. according to ISO 6579-1:2017 [30], and finally, cheese samples were also screened for the enumeration of *Clostridium perfringens* according to ISO 7937:2004 [31].

For each sample, the mean values of viable counts and respective standard deviation error were estimated and expressed as the logarithm of colony forming units per gram of product (log CFUsg^−1^).

### 2.3. DNA Extraction and Genetic Identification of LAB Isolates

The biochemically confirmed LAB isolates obtained from anaerobically incubated MRSA plates were subjected to genetic identification through the amplification and sequencing of the 16S rRNA gene. To that end, the DNA of each isolate was extracted boiling a cell suspension in 50 µL of 0.05 M NaOH solution (Biochem Chemopharma, Cosne sur loire, France) on a heating block (Stuart Scientific, Staffordshire, UK) at 98 °C for 15 min. The lysed suspensions were centrifuged at 10,000× *g* for 5 min, and the lysate (supernatant) was transferred to a new tube and kept in a cooling block prior to the PCR amplification stage. PCR amplification was conducted in a CFX96 thermocycler (BioRad Inc., Hercules, CA, USA) with a final reaction volume of 25 μL, containing 1x iTaq™ Universal SYBR^®^ Green Supermix (2x) (BioRad Inc., Hercules, CA, USA), 200 nM of forward primer 27F (5′-AGAGTTTGATCCTGGCTCAG-3′) (STAB VIDA Lda., Lisboa, Portugal), 200 nM of reverse primer 1492R (5′-GGTTACCTTGTTACGACTT-3′) (STAB VIDA Lda., Lisboa, Portugal) and 2 μL of lysate as template. The amplification protocol consisted of a primary denaturation step at 95 °C for 5 min, followed by 45 cycles of denaturation for 20 s at 95 °C, annealing for 20 s at 55 °C, extension for 90 s at 72 °C and a final extension at 72 °C for 10 min.

The amplified products were purified and Sanger sequenced at STAB VIDA Lda. (Lisboa, Portugal). The resulting chromatograms were analyzed using the UGENE program (version 38.1, Unipro, Novosibirsk, Russia) for the extraction of high-quality fragments, which were aligned and compared against the National Center for Biotechnology Information (NCBI) database using the nucleotide Basic Local Alignment Search Tool (BLAST) function with the default parameters (https://blast.ncbi.nlm.nih.gov/Blast.cgi, accessed on 18 January 2022).

### 2.4. Statistical Analysis

A multiple correspondence analysis (MCA) was applied using the overall frequencies of identified LAB isolates to determine the overall relationship between LAB isolates, food matrix and manufacturing period. This multivariate exploratory method uses nominal or categorical data designed to find correspondence of two or more categorical variables by reviewing the distances between variables, thus allowing a pattern analysis of several categorical dependent variables [32,33]. MCA was calculated using frequency distribution through a Burt table to distribute all variables across the computed dimensions. Variables with the lowest distance were considered as those with the highest degree of similarity in the corresponding dimension. MCA analysis was performed using TIBCO^®^ Statistica^®^ (version 14.0.0, TIBCO Software Inc, Palo Alto, CA, USA). Abundance histograms were performed on GraphPad Prism (version 5.0, GraphPad Prism Software Inc., Boston, MA, USA).

## 3. Results and Discussion

### 3.1. Hygiene and Safety of Serra da Estrela PDO Cheeses

Cheese manufactured from raw milk is a safer food product for human consumption from a microbial perspective than raw milk [8]. In fact, in the United States alone, over 120 food-borne outbreaks due to the consumption of contaminated raw milk were reported in a 20-year period (studied between 1992 and 2012) [34,35]. Additionally, in roughly the same time span (1992 to 2017) in England and Wales, there were over 25 outbreaks reported [36]. This is well above the 60 outbreaks reported worldwide from the consumption of contaminated raw milk cheeses [14,16]. This results from the biochemical reactions taking place during cheese manufacturing, such as nutrient depletion, release of antimicrobial compounds, pH and redox potential reduction, water activity decrease and low temperature ripening stages, which promote a hostile environment for pathogen development [8,11,12]. Nonetheless, spoilage and pathogenic microorganisms can gain access to cheese during manufacturing due to the use of contaminated raw materials, suboptimal fermentations and faulty sanitary or hygiene practices [11,12,14,15]. However, to the extent of our knowledge, no outbreaks related to the consumption of contaminated Serra da Estrela cheeses or other Portuguese PDO cheeses have been reported so far.

To assess the safety and hygiene of Serra da Estrela PDO cheeses, both European law (EC No. 2073/2005) [37] and Portuguese guidelines [38] establishing the microbiological criteria for foodstuffs and ready-to-eat food products were used. Concerning food safety, viable counts of *B. cereus*, coagulase-positive staphylococci and *L. monocytogenes* were not obtained, while *Clostridium perfringens* was found within admissible levels of ≤1.0 log CFUsg^−1^ (Appendix A). Finally, *Salmonella* spp. were not detected in any of the cheese samples (Appendix A). Overall, these results attest the microbial safety of Serra da Estrela PDO cheeses and its manufacturing process, even in the event of occasional colonization of potential hazardous bacteria, such *B. cereus* and coagulase-positive staphylococci, in cardoon and curd samples, respectively. 

Regarding hygienic standards, in the analyzed samples of Serra da Estrela PDO cheeses, *Listeria* spp. ranged from not detected up to 1.94 log CFUsg^−1^, irrespective of the manufacturing period (Appendix A). According to the national guideline, three samples presented unsatisfactory *Listeria* spp. abundance, which is restricted to ≤1.0 log CFUsg^−1^. *Enterobacteriaceae* was found in all the analyzed cheese samples, and it ranged between 3.8 and 5.4 log CFUsg^−1^. *E. coli* comprised about half the counts, with 2.2 to 3.2 log CFUsg^−1^ (Appendix A). The *Enterobacteriaceae* content found in this study was below previous assessments, namely 1 to 3 log [20,21,39,40]. However, *E. coli* incidence increased from ≈10% to 50% of total *Enterobacteriaceae* [20]. Moreover, across the production season, it was possible to observe a slight decrease in overall counts of *Enterobacteriaceae* and *E. coli* load (Figure 2). This behavior was replicated in raw ewe milk samples, and this was previously found to be inversely related to female milk output [41]. Furthermore, *Enterobacteriaceae* transference from cardoon can act as a secondary source of these type of microorganisms since they comprise a significant proportion of its microbiota, which is in agreement with previous microbial characterizations [42,43]. Overall, according to the European law, the Serra da Estrela PDO cheeses analyzed here presented satisfactory *E. coli* parameters, with the exception of two samples that presented counts slightly above the limit threshold of 3.0 log CFUsg^−1^ (Appendix A).

Finally, both European law and Portuguese guidelines are omissive on the threshold values of yeasts and molds in foodstuffs and ready-to-eat food products. The Serra da Estrela cheeses analyzed here presented a load that ranged between 1.0 and 3.9 log CFUsg^−1^, with the highest counts found in late manufacturing (Figure 2 and Appendix A). These figures were generally inferior to the ones reported in previous assessments concerning yeast prevalence [20,21,39,40]. Moreover, seasonal fluctuations also determined that late cheeses were poorer in yeasts when compared to early and mid manufactures, which resulted from a lower lactate availability [20,21]. In this work, however, late cheeses presented the highest yeast figures, possibly due to increased lactic acid availability, as suggested by the higher prevalence of lactococci and lactobacilli in these cheeses when compared to the other two periods (Figure 2 and Figure 3). Overall, yeasts and molds naturally colonize cheese matrices and are frequently added as adjunct cultures in several cheese types [10]. They are usually involved in sugar metabolization, lipolysis and proteolysis activities, thus contributing to its overall organoleptic characteristics [10,11,12]. Furthermore, they can also exert a protective function against colonization of pathogenic and spoilage microorganisms and present probiotic properties [10,11].

In general, these results attest the compliance of Serra da Estrela PDO cheeses with hygienic standards, with exception of occasional issues with *Listeria* spp. and *E. coli* counts above the defined limits. Therefore, sanitary and hygiene practices at the production line should be strengthened, preventing occasional contamination from cross-contamination events and microbial transfer from workers and the factory environment onto cheese.

### 3.2. LAB Microbiota in Serra da Estrela PDO Cheeses

Microbial community composition surveys can be performed using culture-dependent and culture-independent methodologies [7]. The former, despite having a lower resolution power when compared to some culture-independent techniques, such as Next Generation Sequencing, continues to be the gold standard technique for the disclosure and characterization of viable microbiota in any given sample [7].

LAB are ubiquitous bacteria frequently associated with the dairy environment. In cheese production, they are a key part of its microbiota and can act as starter cultures, including strains of *Lactococcus lactis*, *Streptococcus thermophilus*, *Lactobacillus delbrueckii* and *Lactobacillus helveticus*, which are responsible for early lactose metabolization and pH decrease. Their role as secondary or adjunct cultures is also relevant, as heterofermentative lactobacilli, *Leuconostoc* spp., *Pediococcus* spp. and *Enterococcus* spp. strains that in conjugation with other bacteria, yeasts and molds, are ultimately responsible for the acquisition of the intrinsic organoleptic characteristics of each cheese type during the ripening stage [8,9,10,12,44]. As expected, LAB constitute the major fraction of the Serra da Estrela cheese microbiota, ranging from 8.8 to 9.1 log CFUsg^−1^ (Appendix A). Upon analyzing the results, it is apparent that lactococci, lactobacilli and *Leuconostoc* spp. predominate over enterococci strains (Figure 3), which is a microbial behavior also found in previous Serra da Estrela cheese characterizations [39,40,45]. Moreover, three different abundance patterns emerge from the results, as observed in Figure 3. First, viable lactococci grown in M17 and lactobacilli grown in RA presented a tendency to increase across the production season. This is contrary to what was previously reported, where cheeses manufactured in spring presented lower lactococci figures in comparison to autumn and winter productions [20,21]. The number of viable *Leuconostoc* spp. grown in MSE remained roughly unchanged through the production season, whereas viable enterococci grown in SBA decreased during late manufacturing by roughly 1 log CFUsg^−1^ when compared to the values found in early and mid productions. 

Regarding LAB colonization of cardoon samples, it ranged from absent up to 5.8 log CFUsg^−1^, with lactococci emerging as the most predominant LAB group and the winter period yielding, in general, the highest counts (Figure 3; Appendix A). Overall, the cardoon samples analyzed here presented a higher microbial LAB load in comparison to the few previous reports on the subject [42,43].

Concerning raw ewe milk, their LAB content ranged between 5.2 and 6.3 log CFUsg^−1^, with lactococci and lactobacilli predominating over enterococci and *Leuconostoc* spp. strains (Figure 3; Appendix A). These results are in line with the available literature on this type of milk [11]. Moreover, mid-season milks presented the highest LAB abundances in comparison to the other two analyzed periods (Figure 3). This results from the ability of LAB (and especially enterococci) to thrive in high relative humidity and low-temperature environments characteristic of the Serra da Estrela region during winter periods [15,18,46].

Finally, the isolation of viable LAB strains recovered from MRSA plates for each matrix and their genetic identification through 16S Sanger sequencing (Appendix A) allowed the execution of a correspondence analysis (CA) to these data (Figure 4). Generally, the CA discloses possible correspondences between samples and species in a table of counted data, representing it in a reduced ordination space [47]. CA has been widely used in the medical and ecology fields; however, its application to the distribution of microbial abundance along environmental parameters or gradients is not extensive [47,48]. Particularly, its application to food microbiology studies is limited to a few examples [49,50,51]. This analysis reveals relationships between the food matrix and environmental heterogeneity, identifying the main variables that affect bacterial communities within a large set of environmental variables [47]. Thus, it provides a unique perspective in the microbiological characterization of food products, considering location, period, processing, raw materials used, among others.

Upon analyzing the resulting plot, it is possible to identify a group of five species, *Lacticaseibacillus paracasei*, *Leuconostoc mesenteroides*, *Lactococcus lactis*, *Enterococcus durans* and *Enterococcus faecium*, positioned within the area formed by dairy matrices, i.e., milk, curd and cheese and manufacturing periods. This outcome suggests that these strains are essential for Serra da Estrela PDO cheese manufacturing. Concerning *L. paracasei*, it is a heterofermentative LAB found commonly associated with dairy matrices and presents, in this context, interesting proteolytic and lipolytic activities [10,11]. It is active during cheese ripening, and it contributes to cheese flavor and aroma profile development through hydrolysis of bitter peptides and in the release of small peptides and free amino acids [10]. Cheeses made in the presence of this species are characterized by low bitterness and a buttery and sweet aromatic taste [10].

*L. mesenteroides*, also a heterofermentative LAB found associated with dairy matrices, presents an ability to metabolize lactose and citrate, leading to the production of precursors and flavor and aroma compounds [11]. These were found in previous chemical characterizations to be important contributors of the overall organoleptic profile of Serra da Estrela cheeses [18]. Additionally, some *L. mesenteroides* strains can exert significant proteinase, peptidase and lipase activities, as revealed by Macedo and Malcata [52]. The studied strain presented a preference for the release of short- and medium-chain fatty acids similar to the fatty acid profile found in Serra da Estrela cheeses [18,52,53].

*L. lactis* is considered a starter culture, and it is fundamental in the early stages of cheese manufacture [9]. However, and similarly to this study, previous reports on the microbial characterization of Serra da Estrela cheeses found that it can persist throughout ripening stage and reach the finished product [20,23]. Some *L. lactis* strains display relevant proteinase, peptidase and lipase activities, with the ability to metabolize citrate and amino acids into flavor compounds, such alcohols, ketones and aldehydes [11,52]. Moreover, some *L. lactis* strains were found to be able to convert the metabolization products of lactobacilli such ketoacids into aromatic carboxylic acids [10]. In fact, carboxylic acids were found to be the dominant family in the volatile fraction of Serra da Estrela cheeses [45,53].

Enterococci species such as *E. durans* and *E. faecium* are known to be able to reside in milk and other dairy products [54]. They are active during cheese ripening, and they present proteolytic and lipolytic activities that contribute to the development of cheese flavor and aroma due to the production of short- and medium-chain fatty acids, acetaldehyde, acetoin and diacetyl compounds [11,45].

Moreover, some matrix-species-specific associations also emerged from the observation of Figure 4. Centrally on the top of graph is found *Leuconostoc citreum*, which is representative of higher abundance in milk matrices. On the far-left side are found *Streptococcus infantis*, *Enterococcus faecalis*, *Enterococcus lactis* and *Lactococcus garvieae* strains. Their position denotes that these strains were specifically found associated with curd matrices. Likewise, on the far right of the graph are found *Lacticaseibacillus casei*, *Lactiplantibacillus plantarum* and *Latilactobacillus curvatus*, which denotes a closer association with cheese matrices. These strains are heterofermentative lactobacilli usually found associated with cheese microbiota. Their role in cheese flavor and aroma profile development is somewhat similar to that previously described for *L. paracasei* [10,11]. On the bottom of the graph are located *Latilactobacillus graminis*, *Enterococcus gallinarum* and *Enterococcus casseliflavus* strains, which, according to their position, are specifically associated with cardoon matrices. Since they are placed far from any of the analyzed dairy matrices, it is possible to infer that these strains are irrelevant to the manufacturing of Serra da Estrela PDO cheeses.

Similar to the exercise performed above, it is also possible to find some period–species associations. Specifically, lactobacilli strains were more associated with winter/spring manufactures, enterococci were associated with autumn/winter manufactures, and *L. mesenteroides* were positioned equally distant from all periods, indicating its transversal importance across the different manufacturing periods. These observations are in line with the abundance patterns previously disclosed for the different LAB groups that have been studied, attesting the robustness of this statistical analysis (Figure 3).

Finally, Macedo and Malcata [52] suggested a set of *L. mesenteroides*, *L. lactis* and *E. faecium* strains as a putative starter mix for the production of Serra da Estrela cheeses due to their hydrolase potential. Taking in consideration the results of this study, strains of *L. paracasei*, *E. durans*, *L. curvatus*, *L. casei* and *L. plantarum* could be considered putative candidates.

## 4. Conclusions

In this work, samples of Serra da Estrela PDO cheese, curd and raw materials, ewe milk and vegetal rennet (dried flowers of *Cynara cardunculus* L.) gathered across the manufacturing season of 2018/2019 were microbiologically characterized concerning their content of LAB and hygienic and food safety indicators. The assessment of hygienic and food safety microbiological indicators revealed that Serra da Estrela cheeses are a safe food product. Occasionally, some cheese samples presented unsatisfactory coliform and *Listeria* spp. parameters.

Concerning the content in lactic acid bacteria, this study showed that cheese presented the highest abundance, followed by curd, ewe milk and finally cardoon. The correspondence analysis performed on LAB isolates revealed that *L. paracasei*, *L. lactis*, *E. durans*, *E. faecium* and *L. mesenteroides* strains are fundamental in Serra da Estrela cheese production, as they are present in milk, curd and cheese matrices, irrespective of the manufacturing period of analysis. Furthermore, lactobacilli were more closely associated with mid and late manufacturing periods, while enterococci were preferably found in early and mid manufacturing periods.

Overall, this study contributes an updated microbiological characterization and safety assessment of the most important and recognizable traditional Portuguese cheese product, whose information on current manufacturing practices of the 21st century were lacking.

## Figures and Tables

**Figure 1 foods-12-02008-f001:**
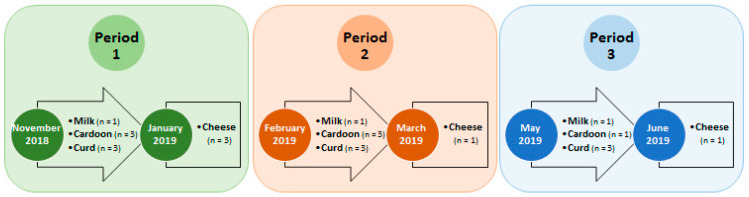
Sampling strategy followed in this study for the microbial characterization of ewe raw milk, cardoon, curd and Serra da Estrela PDO cheese. The letter n refers to the number of samples analyzed for the corresponding sample in that specific period.

**Figure 2 foods-12-02008-f002:**
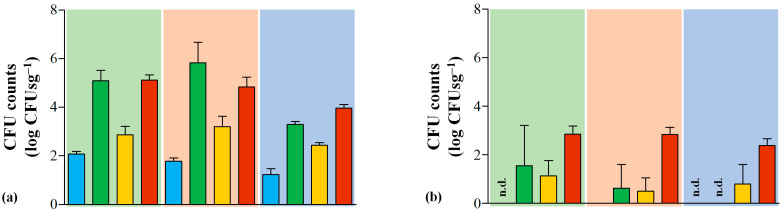
Abundance histogram with standard deviation error bars of (**a**) *Enterobacteriaceae* (according to ISO 21528-2:2017); (**b**) *E. coli* (according to ISO 16649-2:2001); (**c**) molds and (**d**) yeasts (according to ISO 21527-1:2008) found in raw ewe milk (blue), cardoon (green), curd (yellow) and Serra da Estrela PDO cheese (red) matrices, expressed in logarithm of colony forming units per gram of cheese (log CFUsg^−1^). The background color is representative of a specific manufacturing period: November–January (green); February–March (orange); May–June (blue). n.d. refers to non-observed counts for a given matrix in that corresponding manufacturing period.

**Figure 3 foods-12-02008-f003:**
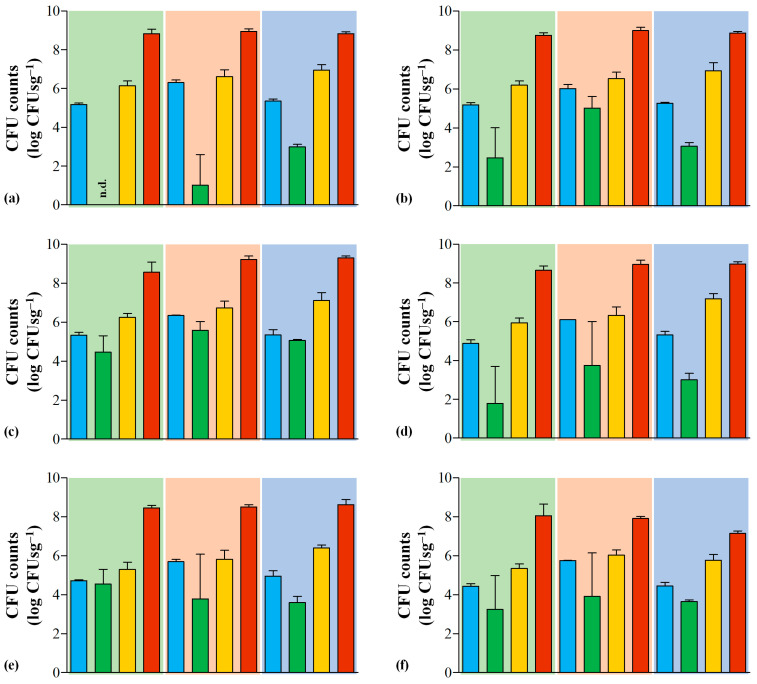
Abundance histogram with standard deviation error bars of presumptive (**a**) lactic acid bacteria—aerobic counts; (**b**) lactic acid bacteria—anaerobic counts; (**c**) lactococci; (**d**) lactobacilli; (**e**) *Leuconostoc* spp. and (**f**) enterococci found in raw ewe milk (blue), cardoon (green), curd (yellow) and Serra da Estrela PDO cheese (red) matrices expressed in logarithm of colony forming units per gram of cheese (log CFUsg^−1^). The background color is representative of a specific manufacturing period: November–January (green); February–March (orange) and May–June (blue). n.d. refers to non-observed counts in cardoon samples from the November–January period.

**Figure 4 foods-12-02008-f004:**
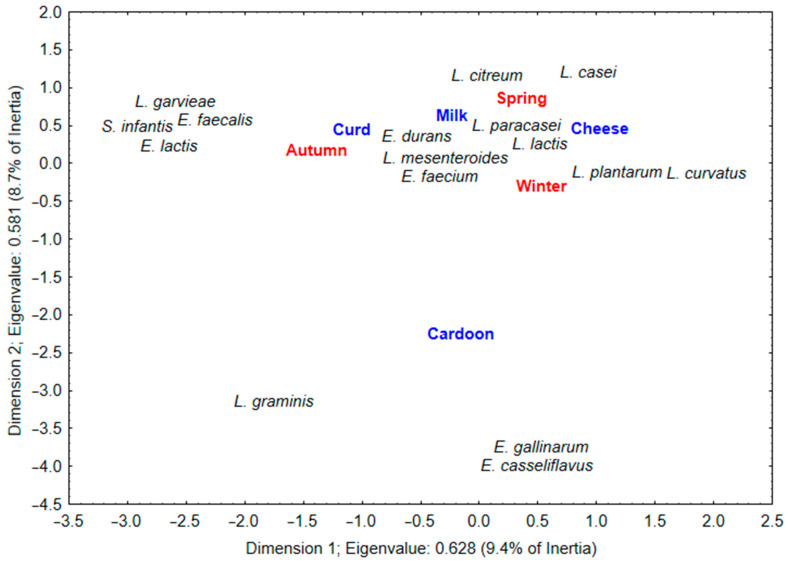
Multiple correspondence analysis plot displaying the association between matrix (in red) and manufacturing period (in blue) with identified viable lactic acid bacteria grown on Man, Rogosa e Sharpe Agar in Serra da Estrela PDO cheese and raw materials.

## Data Availability

The data used to support the findings of this study can be made available by the corresponding author upon request.

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
