# Peer review of "Microbiological Characterization of Protected Designation of Origin Serra da Estrela Cheese"

_foods, 2023, doi:10.3390/foods12102008_

Round 1

Reviewer 1 Report

Dear Authors, 

I had the opportunity to review your paper about the microbiological profile of cheese samples with a protected designation of origin. After reading and reviewing this paper, I think that this study has the potential to be published, but must be revised in some particular spots in the manuscript.

First of all, the use of abbreviations in the title gives a wrong impression and does not emphasize the special aspect of this work. Accordingly, I suggest that they put the full name anyway, avoiding the abbreviation in the title.

Throughout the text, the units of measure should be standardized, especially for the number of microorganisms. Check whether they should be written as CFUg-1 or CFU/g.

The abstract is too long, shorten it to about 200 words. Throw out unnecessary technical data and better emphasize the goal of the work.

I suggest that you insert and adjust the second paragraph from the abstract at the end of the Introduction. Other than that, I have no objections to the literature review.

I believe that the ISO standards used are also references, and should be included as such in the List of References and numbered in the work.

Table 1 gives the impression of sloppiness and is very unreadable. Try to present the results in a better way.

Same comment for Table 2.

The biggest complaint concerns the Multiple correspondence analysis plot, which is not adequately presented or explained in the text. I would ask that this entire section be revised, data added and commented on in a more appropriate way compared to the available analysis.

Author Response

Dear all, 

In the word document attached to this communication we present, point by point, our answers to the comments made by Reviewer nr. 1 to our work.

Regards

Rui

Reviewer 2 Report

In the submitted manuscript, the authors have characterized the microbial composition of Serra da Estrela PDO cheeses and its raw materials across the three different campaigns.  Through the result, it has been clarified that when some kinds of LAB species and/or some pathogenic bacteria are predominant at fermenting stages.  The results are of interest and provide updated safety assessment in manufacturing cheese products.

Although there are some typographical errors in the submitted manuscript, the present study seems to be performed with appropriate methods, thus the manuscript may satisfy the minimal criteria for publishing in Foodsas scientific paper.  However, there are some concerns for publication at present form as follows:

1. The authors surely have indicated profiles of pathogenic and LAB species on different types of materials at different seasons.  However, there are no details on substances contained in those materials that may contribute to flavor, such as sourness, bitterness, aroma, and good tastes. (Further, some beneficial ones have health-promoting effects).  I suggest that the authors should add that information into the manuscript for ease of understanding the aims of the present study to readers.

2. Some species of Yeasts and Molds are also known as beneficial (probiotic) species.  How would the authors like to add some discussion on those in the manuscript?

3. The authors use colony-culture method instead of NGS to analyze the composition of the materials, because they may want to evaluate living microorganisms.  I recommend to they emphasis the description on that.

Author Response

Dear all, 

In the word document attached to this communication we present, point by point, our answers to the comments made by Reviewer nr. 2 to our work.

Regards

Rui

Round 2

Reviewer 1 Report

Dear Authors,

I am especially pleased with your effort and all the changes you have made in your text. I am going to suggest the acceptance of this paper.